# OmniStitch: Depth-Aware Stitching Framework for Omnidirectional Vision with Multiple Cameras

Omitted for Double-blind Review

## ABSTRACT

Omnidirectional vision systems provide a 360-degree panoramic view, enabling full environmental awareness in various fields, such as Advanced Driver Assistance Systems (ADAS) and Virtual Reality (VR). Existing omnidirectional stitching methods rely on a single specialized 360-degree camera. However, due to hardware limitations such as high mounting heights and blind spots, adapting these methods into vehicles of varying sizes and geometries is challenging. These challenges include limited generalizability due to the reliance on predefined stitching regions for fixed camera arrays, performance degradation from distance parallax leading to large depth differences, and the absence of suitable datasets with ground truth for multi-camera omnidirectional systems. To overcome these challenges, we propose a novel omnidirectional stitching framework and publicly available dataset tailored for varying distance scenarios with multiple cameras. The framework, referred to as , consists of a Stitching Region Maximisation (SRM) module for automatic adaptation to different vehicles with multiple cameras and a Depth-Aware Stitching (DAS) module to handle depth differences caused by distance parallax between cameras. In addition, we create and release an omnidirectional stitching dataset, called which provides ground truth images that maintain the perspective of the 360-degree FOV, specifically designed for vehicle-agnostic systems. Extensive evaluations of this dataset demonstrate that our framework outperforms state-of-the-art stitching models, especially in handling varying distance parallax. The proposed dataset and code are publicly available in URL.

## CCS CONCEPTS

• **Computing methodologies** → **Image processing**; **Virtual reality**; *Reconstruction*.

## KEYWORDS

Omnidirectional vision, Image stitching, Omnidirectional view dataset

**ACM Reference Format:**
Omitted for Double-blind Review. 2018. OmniStitch: Depth-Aware Stitching Framework for Omnidirectional Vision with Multiple Cameras. In *Proceedings of Make sure to enter the correct conference title from your rights confirmation emai (Conference acronym 'XX)*. ACM, New York, NY, USA, 10 pages. https://doi.org/XXXXXXX.XXXXXXX

## 1 INTRODUCTION

Omnidirectional vision systems are becoming increasingly important in a wide range of applications, from advanced driver-assistant systems (ADAS), Virtual Reality (VR) to Unmanned Robotics and Mars exploration [8, 43, 44, 46], providing essential 360-degree environmental awareness. Traditional image stitching methods for omnidirectional vision [17, 30, 32, 51, 55] that rely on specialized 360° cameras (*e.g.*, Samsung Gear 360[1] and Insta360[2]) mainly focus on single camera settings with multiple wide-angle lenses and synthesize the images from difference views overcoming large angular parallax. However, the single-camera setting is not suitable for general vehicle applications such as autonomous driving [14, 16, 47, 48, 58] due to high installation requirements [21, 26] and blind spots [15, 45], which limit their effectiveness across different vehicle sizes and designs. To provide all-around 360° coverage for vehicle-agnostic omnidirectional vision system (VA-OVS), image stitching techniques with multiple camera settings should be adopted as illustrated in Figure 1.

Addressing the technical challenges of omnidirectional image stitching in VA-OVS scenarios presents several technical difficulties. First, the existing image stitching methods are limited to inputs from 360° cameras [17, 24, 35, 51] or specific camera arrays designed for predefined [23, 56], manually crafted stitching regions. When adapting to different camera settings, manually aligning the images, transforming the viewpoint, and selecting a well-matched stitching region require redundant human efforts. Second, these methods struggle with distance parallax between different camera inputs. Our observation indicates that increased distance parallax leads to diminished stitching performance, as illustrated in Figure 1. This degradation is mainly due to the extreme depth differences across images, which prior methods overlook, resulting in errors such as ghosting and misalignment artifacts. Third, there are no data sets with ground truths for omnidirectional image stitching with dynamic distance parallax that can be applied to VA-OVS scenarios.

To address these challenges, we propose a novel omnidirectional stitching framework, called OmniStitch for dynamic distance consideration with multiple cameras. OmniStitch has two main modules: Stitching Region Maximization (SRM) and Depth-Aware Stitching (DAS). The SRM module solves the stitching region decision difficulty by simplifying the image alignment and viewpoint transformation process by statically matching half of the image for each stitching input pair. By maximizing the candidate size of image stitching, the framework can learn dynamic distance parallax between cameras and eliminate the need for pre-processing and post-processing. After the SRM process, the DAS module transforms image pairs by considering large depth differences between image pixels and extracted context features to generate a seamless

---

[1]https://www.samsung.com/sec/gear/360/
[2]https://www.insta360.com/

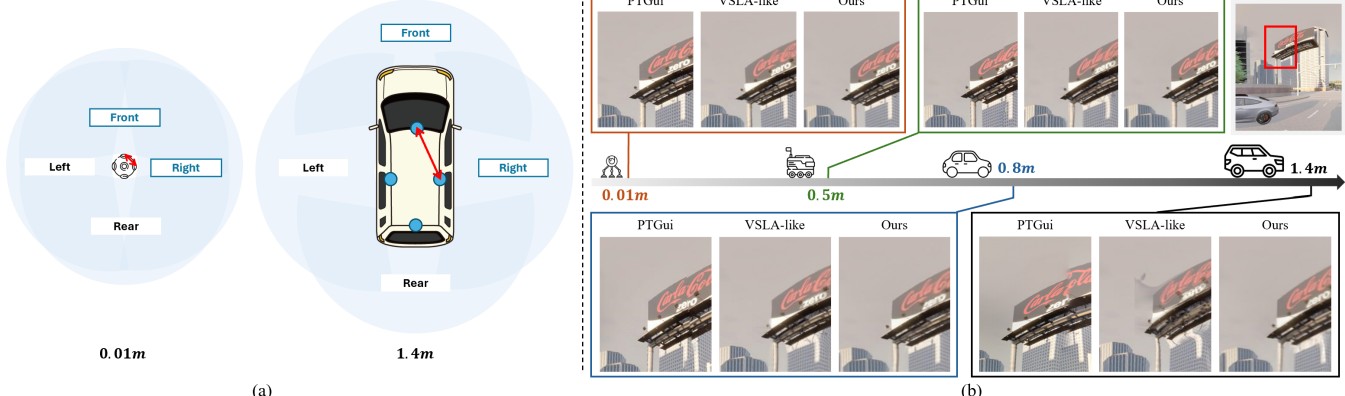

**Figure 1: Examples of Vehicle Agnostic Omnidirectional Vision System (VA-OVS) (a): Inputs with dynamic distance for four-way camera configuration. (b): Comparison of the outputs of the image stitching methods with the ground truth image.**

360-degree image. It enhances traditional flow estimation by incorporating additional depth information computed from 3D feature matching, and warps the images with learnable weights at the pixel level to consider large depth differences between the images.

For the challenge of dataset scarcity in the field of omnidirectional vision, we release the General-purpose Vehicle's 360-degree (GV360) dataset, leveraging the CARLA [10] simulator to generate ground-truth images in a virtual environment. This dataset is designed with data captured from four-way multiple cameras, incorporating a wide range of dynamic distance parallax inputs and preserving a 360° FOV perspective. The use of virtual environments overcomes the limitations associated with real-world data collection, broadening the scope for future research in vehicle-agnostic omnidirectional vision systems (VA-OVS). Experimental results using the GV360 demonstrate that our framework outperforms the state-of-the-art stitching model by an average of 18.5% of PSNR and 18.0% of LPIPS for all tested distance parallax scenarios. Furthermore, our framework demonstrates superior performance on unseen datasets in qualitative results.

The main contributions are summarized as follows:

- We propose a novel end-to-end omnidirectional image stitching framework for dynamic distance parallax in multiple camera settings. It enhances generalization of stitching framework to multi-camera Vehicle-Agnostic Omnidirectional Vision System (VA-OVS) scenarios by effectively eliminating the need for a hand-crafted stitching region decision process.
- Our framework outperforms the state-of-the-art stitching methods, regardless of variations in distance parallax between cameras. The SRM module simplifies the manual process of identifying stitching regions for multiple cameras, and the DAS module effectively handles the artifacts caused by distance parallax between cameras.
- To tackle the dataset sparsity problem in omnidirectional stitching fields, we release a dataset for omnidirectional vision stitching, referred to as GV360. This dataset includes ground-truth images that capture 360-degree perspectives and simulate varying distance parallax scenarios across different vehicle types.

## 2 RELATED WORK

### 2.1 Unidirectional Stitching

Most image stitching models are designed for unidirectional vision applications, which generate a panoramic image from the viewpoint of a reference image. Unidirectional stitching methods merge two images by warping one towards the other to seamlessly stitch boundary regions [4, 6, 11, 13, 18, 25, 27, 28, 31, 36, 40, 54, 59?, 60]. Various techniques were proposed to divide the image into sub-regions and estimate homography for each region, using ground and distance planes [13], grid units [28, 59, 60], triangles [27, 36], and superpixels [25]. Other enhancements have included maintaining image structure through local or global similarity transformations [4, 6, 11, 18, 31] and leveraging deep learning-based stitching models [40] for improved geometric integrity. However, these methods still suffer from projective distortion and misalignment artifacts in the case of wide parallax, due to the homography transformation process, which distorts the target image to fit the viewpoint of a reference image. Although VSLA [23] proposes to use bi-directional flow estimation to synthesize an intermediate view for wide parallax between cameras, the need for specific camera arrays and extensive pre-processing reveals the limited adaptability to various vehicles in VA-OVS scenarios.

### 2.2 Omnidirectional Stitching

Omnidirectional stitching combines images from all directions to create a 360-degree view, providing a perspective view of spherical maps such as the equirectangular projection (ERP) format. It is commonly used with 360-degree cameras [17, 51] or circular rig cameras [2, 35], which can produce 360-degree images by synthesizing each viewpoint. Traditional stitching methods for omnidirectional vision rely on mathematical approaches, focusing on geometric calibration through feature points [7, 17, 32, 55], calibration boxes[57], and depth map integration for seamless multi-camera setting [2], as well as advanced optical flow methods [35]. On the other hand, deep-learning approaches can significantly enhance the performance of omnidirectional stitching, such as attention mechanisms [12, 30] for matching SR to human visual perceptions and weakly-supervised

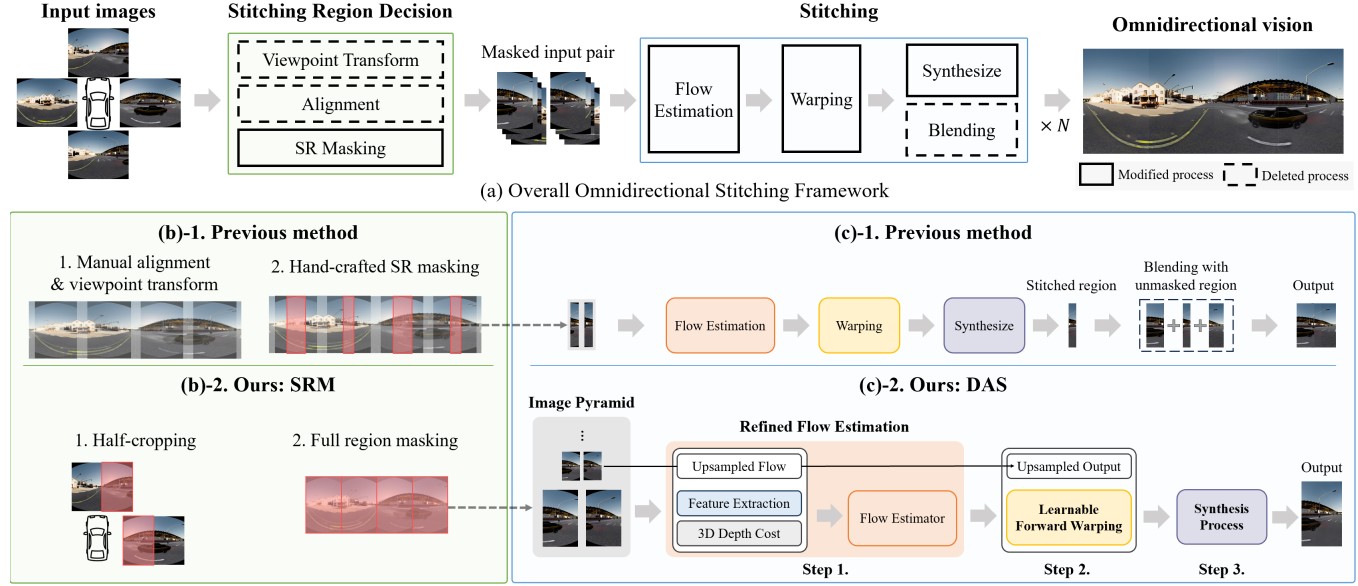

(a) Overall Omnidirectional Stitching Framework

(b) Stitching Region Decision

(c) Overview of Stitching process

**Figure 2: Overview of image stitching framework. In (a) and (c), the borderline boxes represent the modified process, and the solid lines represent the deleted process from the previous stitching framework thanks to the stitching region maximization (SRM) module. (b) and (c) are divided into the previous method and our proposed method. In (c), the orange, yellow, and purple boxes mean each of the consequent steps.**

learning [51] for real-world ground truth image training. However, despite their efforts for wide parallax, they only support fixed camera formations (*e.g.*, dual 195° FOV lenses or quad 190° FOV lenses) with a single 360° camera, and do not consider dynamic distance parallax in scenarios requiring flexible camera positioning.

## 2.3 Image Stitching Datasets

Datasets are crucial for the development of image stitching methods, particularly those that require ground truth (GT) or deal with parallax. View-free [37], Stitched MS-COCO [41], and DIR-D [39] provide GT for unidirectional stitching but lack parallax consideration. In contrast, datasets for large parallax such as VSLA [23], UDIS-D [38] and PDIS [22] suffer from restricted access or lack of GT, limiting their applicability. For omnidirectional stitching, CROSS [29] and WSSN [51] provide accessible GT but are confined to single camera systems, restricting the versatility of camera configurations. The GV360 dataset fills these gaps by supporting multiple camera settings for VA-OVS using virtual environments to ensure adaptability across different scenarios, thus broadening the scope for stitching algorithm development.

## 3 DEPTH-AWARE OMNIDIRECTIONAL STITCHING FRAMEWORK

### 3.1 Overview

Our goal is to integrate images from multiple cameras into a single omnidirectional image, accommodating variations in distance parallax that arise due to the diverse sizes and geometries of vehicles. In this work, we identify two requirements for VA-OVS scenarios:

1) the implementation of a multi-camera setting and 2) adaptability to dynamic distance between cameras depending on diverse vehicle types. This approach assumes a configuration with four surrounding cameras mounted on the vehicle's body. This is the minimum number of cameras in a typical scenario with multiple cameras [15, 47, 48, 58], arranged at 90° angles between them, providing an omnidirectional configuration ideal for vehicles of various shapes and sizes. We consider a dynamic distance parallax scenario where the distance between two cameras can vary from 0.01m to 1.4m, which is suitable for various vehicles.

The overall process of the omnidirectional stitching framework is illustrated in Figure 2. Given the input $\{IMG_i\}_{i=1}^N$, where $N$ is the number of image inputs (*i.e.*, $N = 4$), the general framework first identifies the optimal stitching regions for each pair of adjacent images. These regions are denoted as $\{IMG_{(i,r)}, IMG_{(i+1,l)}\}_{i=1}^N$, where $r$ and $l$ indicate the identified right and left parts of the images, respectively. These paired images undergo bidirectional flow estimation and pixel-level warping during the stitching phase, resulting in the output $\{O_i\}_{i=1}^N$, where each output

$$O_i = Stitching(IMG_{(i,r)}, IMG_{(i+1,l)})$$

is a segment of the panoramic image created by the stitching process. We simplify the notation of stitching region pairs from $\{IMG_{(i,r)}, IMG_{(i+1,l)}\}$ to $\{I_L, I_R\}$.

To ensure automatic adaptation of stitching region pairs $\{I_L, I_R\}$ to input images with dynamic distance parallax caused by multiple camera settings, we suggest using the Stitching Region Maximization (SRM) module instead of the manual process before stitching. Furthermore, the Depth-Aware Stitching (DAS) module is proposed

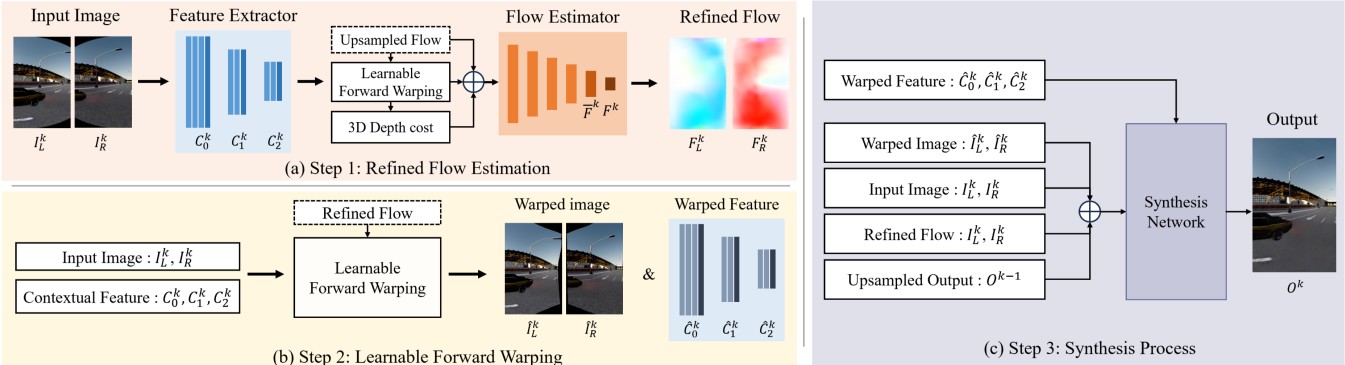

(a) Step 1: Refined Flow Estimation

(b) Step 2: Learnable Forward Warping

(c) Step 3: Synthesis Process

**Figure 3: The step-wise details of the DAS module. The refined flow estimation process (Step 1) from input images to refined flow is illustrated in (a), while (b) and (c) present the input and output configurations for the learnable forward warping and synthesis process, respectively.**

to address artifacts of synthesized outputs $O_i$ caused by depth differences by utilizing additional depth information and an advanced warping network.

## 3.2 Stitching Region Maximization Module

In previous approaches, the SR decision for each input pair typically involves manually transforming the viewpoint and aligning the images or using automatic algorithms such as PTGui[3]. In this case, the region near the seamline of the aligned images is defined as the SR pairs. Only this region will be masked and stitched, followed by a post-processing step such as blending with other unmasked regions. Conversely, we implement half-cropping and full-region matching for each pair $I_L, I_R$, thereby maximizing the stitching region. This technique simplifies the SR decision process by eliminating the need for manual pre-processing (*e.g.*, image alignment, viewpoint transformation, and stitching region decision) and post-processing (*e.g.*, blending with unmasked regions), facilitating adaptation to various camera formations with dynamic distance.

Although the SRM allows for adaptation to various vehicles with multiple camera settings, its performance may decrease when adjusting to situations with large distance parallax. This is due to the increased candidate size for flow estimation, which can confuse the pixel-wise warping process. To handle the increased number of candidates resulting from SRM, we use a cascaded refinement strategy. It is based on the up-sampled features, flow, and synthesis results, which are fed into the next level of an image pyramid structure inspired by the frame interpolation network [19, 49, 61]. This structure organizes the stitching process into multiple levels, with each level $k$ of the total $K$ levels refining the input by incorporating up-sampled output from the previous level. This iterative refinement process ensures a more accurate and robust output.

## 3.3 Depth-Aware Stitching Module

While addressing the challenge of stitching region decisions to handle variations in distance parallax, it was observed that stitching performance decreases in scenarios with large distance parallax, as shown in Figure 1. Taking inspiration from the task of Video

Frame Interpolation (VFI) [9] tasks, which manage large motions and depth differences – challenges similar to those in large distance parallax stitching – we develop the Depth-Aware Stitching (DAS) module. The DAS module incorporates a methodological enhancement inspired by video frame interpolation models [19, 42, 52], which are tolerant of large motion. It modifies each step of the traditional image stitching process to address such challenges related to dynamic distance parallax. The module consists of three main steps: Refined Flow Estimation (Step 1), Learnable Forward Warping (Step 2), and Synthesis Network (Step 3). Each step integrates elements of the UPR-Net [19] structure adapted to the specific requirements of VA-OVS scenarios, with further details provided in Figure 3.

*3.3.1 Step 1: Refined Flow Estimation.* Traditionally, pixel-level flow features are extracted directly from the input pair's stitching region $\{I_L, I_R\}$ using a basic flow estimator $\mathcal{F}(\cdot)$ composed of stacked convolution layers. At each pyramid level $k$, traditional methods utilize outputs from the fifth and sixth layers of a stacked convolution neural network (CNN) to define flow features $\bar{F}^k$ and flow $F^k$ respectively as:

$$\bar{F}^k = CNN_5(I^k), F^k = CNN_6(I^k)$$

with bi-directional flow estimation.

To enhance flow estimation in the context of large distance parallax, our framework inspired by frame interpolation methods [19, 20], leveraging multi-scale feature extraction, up-sampled flow from previous pyramid level, and 3D depth costs. This approach mitigates flow estimation errors caused by large candidate sizes from stitching region maximization using up-sampled flow and brightness discrepancies caused by large parallax. Incorporating high-level contextual information via multi-scale features effectively reduces these impacts, providing additional spherical depth information and considering both explicit and implicit structural information.

Specifically, the feature extractor consists of three multi-scale stages, with each stage comprising 3, 4, and 4 convolution layers, respectively, which process both left and right images to produce context features $\{C^k_{(L,0)}, C^k_{(R,0)}\}$, $\{C^k_{(L,1)}, C^k_{(R,1)}\}$, and $\{C^k_{(L,2)}, C^k_{(R,2)}\}$ at increasing channel depths of 24, 48, and 96. These features are

³https://ptgui.com

**Table 1: Comparison of datasets for wide parallax stitching. It includes camera settings, input types, camera formats, distance parallax, availability, and output field-of-view. The camera format contains the degree of utilized lenses.**

| Model | Camera setting | Input type | Camera format | Distance parallax | Availability | Output FOV |
|---|---|---|---|---|---|---|
| CROSS[29] | Single 360° camera | Dual 180° | Fisheye (195°) | 0.01 m | Y | 360° |
| WSSN[51] | Single 360° camera | Triple 120° | Fisheye (195°) | 0.01 m | Y | 360° |
| Quad-fisheye[7] | Single 360° camera | Quadratic 90° | Fisheye (190°) | 0.01 m | N | 360° |
| VSLA[23] | Multicamera | Triple 45° | Rectilinear (120°) | 1.28 m | N | 210° |
| GV360 (Ours) | Multicamera | Quadratic 90° | Fisheye (185°) | 0.01 m - 1.4 m | Y | 360° |

then used to compute the 3D depth cost, accurately match features between aligned images, and adjust for spherical depth differences. The refined flow encoder integrates these depth costs with up-sampled flow, formulated as:

$$F^k = \mathcal{F}(F^{k-1} \oplus \bar{F}^{k-1} \oplus Cost(C_2^k) \oplus \widehat{C}_2^k)$$

with 3D depth cost $Cost(C_2^k)$ and warped feature $\widehat{C}_2^k$ related to the current feature level.

*3.3.2 Step 2: Learnable Forward Warping.* Given the input image pair $I_L^k, I_R^k$ and optical flow $F_L^k, F_R^k$, the traditional flow-based warping methods typically use mathematical backward warping to produce the warped image pair:

$$\{\widehat{I}_L^k, \widehat{I}_R^k\} = \mathcal{W}(I_L^k, I_R^k, F_L^k, F_R^k)$$

However, these methods result in issues such as object occlusion or duplication in scenarios with large depth differences. To address these challenges effectively, we employ a learnable forward warping technique inspired by the softmax splatting method [42], which enhances the handling of depth differences and reduces artifacts in the warped images. Specifically, given the input image pair $\{I_L^k, I_R^k\}$ and refined flow $\{F_L^k, F_R^k\}$, the warped left image is defined by the equation:

$$\widehat{I}_L^k = \overrightarrow{w}(I_L^k, F_L^{k-1}/2, Z)$$

where $\overrightarrow{w}$ represents the forward warping function. It utilizes half of the up-sampled flow $F_L^{k-1}$ to find intermediate pixel positions between the left and right images, while a depth metric $Z$ aids in accurate pixel mapping. Adopting the softmax splatting approach, the depth metric $Z$ is used to assign appropriate weights to pixels within $I_L^k$ for their projection onto corresponding target pixels in $I_R^k$, expressed as:

$$Z = \lambda \cdot v(C_L^k, (\| C_L^k - \overleftarrow{w}(C_R^k, F_R/2) \|_1)$$

where the weighting procedure is derived from backward warping $\overleftarrow{w}$ and a scaling map $\lambda$, with $v$ representing a U-Net for training.

The core of our learnable forward warping technique lies in its capacity to adaptively determine the source of pixels, especially in scenarios where multiple pixels may target the exact location of the image. This process is made differentiable, allowing for effective learning and optimization. This warping process is also used for feature warping $\widehat{C}^k$ during the flow estimation process as outlined in Step 1.

*3.3.3 Step 3: Synthesis Process.* Finally, recursively warped images $\widehat{I}_L^k$ and $\widehat{I}_R^k$ are combined using an encoder-decoder structure based on the U-Net framework [19, 20, 61] to generate an intermediate view between two images. Given synthesis network inputs, which include the current stitching sources $I^k, \widehat{I}^k, F^k$, and the recursive output $O^{k-1}$ from the prior pyramid level, the stitched image for each pyramid level $k$ is formulated as:

$$O_i^K = \frac{M_L^k \odot \widehat{I}_L^k + M_R^k \odot \widehat{I}_R^k}{M_L^k + M_R^k} + \Delta^k$$

where $M_L^k$ and $M_R^k$ are the masking maps generated by the synthesis network, located in the fourth and fifth channels, respectively, and $\Delta^k$ represents a residual map contained within the first three channels. The final omnidirectional image $O$ is obtained by smoothly concatenating the stitched segments $\{O_i\}_{i=1}^N$.

## 3.4 Training Loss

OmniStitch framework aims to optimize the parameters $\theta$ across the feature encoder, flow encoder $\mathcal{F}$, warping network $\mathcal{W}$, and stitching network $\mathcal{S}$, jointly minimizing the three distinct losses during the training phase. These include the Charbonnier [5] and VGG loss [50] for pixel-level accuracy and assessing perceptual quality, respectively. In addition, the Census loss $\mathcal{L}_{cen}$, known for its effectiveness in handling object occlusion [34], is used to consider large depth differences between two images. The overall training loss is expressed as:

$$\mathcal{L}_{train} = \mathcal{L}_{Char} + \alpha \mathcal{L}_{vgg} + \beta \mathcal{L}_{cen}$$

where $\alpha$ and $\beta$ are calibrating each loss term's scale.

## 4 GV360 DATASETS

Our goal is to facilitate supervised learning for omnidirectional image stitching to produce precise 360°Equirectangular Projection (ERP) images, addressing the need for accurate ground-truth (GT) data and adaptability to multi-camera configurations in typical vehicles that closely mirror real-world conditions. As shown in Table 1, although several datasets exist for wide parallax image stitching [7, 23], there are still limitations in terms of public availability and provision of accurate GT images required for training supervised image stitching models. Certain omnidirectional stitching datasets [29, 51] provide GT images captured with specialized cameras, but these do not align with the requirements of the VA-OVS scenarios. To address these issues, we introduce a new method for generating datasets that can be used in multi-camera settings and

**Table 2: Quantitative result of the GV360 dataset. The bold represents the best performance and the underline represents the secondary performance.**

| Approach | Method | GV360 dataset | | | | | | | | |
| --- | --- | --- | --- | --- | --- | --- | --- | --- | --- | --- |
| | | (a) 0.01 m | | | (b) 0.8 m | | | (c) 1.4 m | | |
| | | PSNR(↑) | SSIM(↑) | LPIPS(↓) | PSNR(↑) | SSIM(↑) | LPIPS(↓) | PSNR(↑) | SSIM(↑) | LPIPS(↓) |
| Unidirectional | APAP | 11.586 | 0.087 | 0.310 | 11.362 | 0.073 | 0.350 | 10.688 | 0.051 | 0.411 |
| | UDIS++ | 13.435 | 0.129 | 0.215 | 12.724 | 0.104 | 0.266 | 10.903 | 0.066 | 0.375 |
| | LPC | 13.531 | 0.111 | 0.193 | 13.416 | 0.094 | 0.216 | 12.831 | 0.072 | 0.233 |
| | VSLA-like | 23.634 | 0.815 | 0.143 | 23.398 | 0.780 | 0.206 | 23.857 | 0.779 | 0.224 |
| Omnidirectional | Samsung Gear 360 | 14.772 | 0.079 | 0.241 | 14.430 | 0.073 | 0.256 | 14.341 | 0.071 | 0.262 |
| | PTGui | 18.885 | 0.227 | 0.229 | 14.832 | 0.153 | 0.284 | 12.597 | 0.061 | 0.327 |
| | OmniStitch | **29.146** | **0.911** | **0.108** | **27.147** | **0.871** | **0.169** | **27.710** | **0.870** | **0.192** |

are not specific to a particular vehicle. We present the General-Purpose Vehicle's 360-degree (GV360) dataset generated by this method. This GV360 dataset is created using the CARLA simulator, an open-source platform that can simulate various traffic and weather conditions and camera configurations.

We render images using fisheye lenses with a wide 185-degree field of view. To satisfy vehicle-agnostic quadratic multi-camera settings [15, 47, 48, 58], we attach a total of four cameras to the upper side (U), downside (D), left (L), and right (R) of each vehicle as illustrated in Figure 1. The distance between cameras varied from 0.01m to 1.4m, adjusting to the various shapes and sizes of the vehicles. The four ground-truth images are aligned for different input pairings: front-right (RU), right-back (RD), back-left (LD), and left-front (LU). These GT images are derived by collecting the center pixels from camera inputs and continuously placing them along an ellipse. This approach minimizes distortion and closely replicates a real-world panoramic view for each side. We place 29 cameras on each of the four intermediate sides to ensure comprehensive coverage. We then un-warp the camera inputs, crop the pixels closest to the center, and align and blend the cropped pixels to reduce camera distortion, creating what we consider to be the ground truth image. The camera arrangement in this setup is inspired by the traditional rigged camera systems (*e.g.*, Google Jump [1]), where cameras are placed at 3° intervals from the center to produce a high-fidelity intermediate image. The dataset consists of 12,724 training samples and 1,284 test samples. The proposed dataset and codes are made publicly available via a provided URL.

## 5 EXPERIMENTS

### 5.1 Implementation Details

The weights in the training loss (3.4) are set to be $\alpha = 1$ and $\beta = 0.01$. The training process spans 300,000 iterations with a batch size of 10, utilizing the AdamW optimizer [33] with a weight decay of $10^{-4}$. The learning rate is gradually reduced through cosine annealing from $10^{-4}$ to $10^{-5}$. The implementation is based on Pytorch 2.1.0 with CUDA 12.1 on Ubuntu 18.04, and the training is performed on two GPUs (NVIDIA RTX 4090 and RTX A6000). To augment

the training data, we randomly crop 480 × 480 patches from the original data and apply random rotations, vertical flips, and channel inversion. For the image pyramid architecture, we decided to use a pyramid level of 4 and skip the flow estimation in the last level of the pyramid, instead using the cascaded flow empirically.

### 5.2 Baselines

To validate the performance of models in omnidirectional image stitching for general vehicle scenarios, we examine two categories of state-of-the-art models. First, we focus on undirectional stitching methods, which predominantly handle pair-wise or linear arrays of image inputs.

- **APAP** [59] is a widely adopted stitching method, recognized for an As-Projective-As-Possible approach. It blends non-projective deviations into a globally projective framework.
- **LPC** [18] emerges as a novel solution aimed to overcome the wide parallax issue. It leverages co-planar region matching alongside global colinear structures, ensuring the preservation of geometrical integrity on both local and global scales.
- **UDIS++** [40] is a parallax-tolerant, unsupervised deep-learning model for image stitching that handles large parallax challenges and improves generalization across different datasets and resolutions.
- **VSLA-like** [23] is a representative stitching model that utilizes partial stitching regions. Since the original code is not open source, we built our own version based on the descriptions in the paper. We trained it with GV360 dataset and set the stitching regions to the same proportions as in the original model (40% of the input image).

Second, we explore omnidirectional stitching methods designed to produce 360° panoramic images.

- **Samsung Gear 360** [17] is a commercial stitching solution using dual fisheye lenses. Due to the limited reproducibility of many stitching methods, it is a benchmark for generating omnidirectional images.
- **PTGui** is a widely used commercial software for automatic stitching that employs feature points for image alignment,

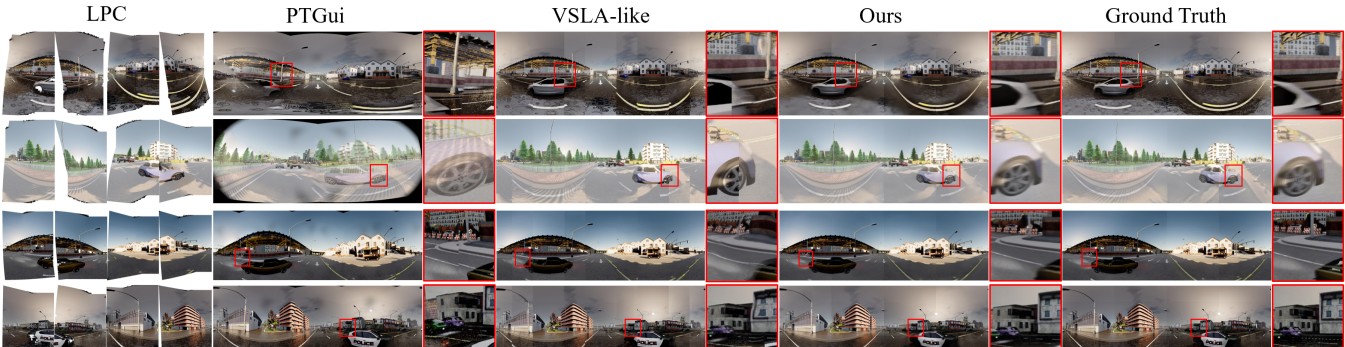

**Figure 4: Qualitative results on GV360 dataset. Distance parallax = 1.4 m (1 and 2 rows), and 0.8 m (3 and 4 rows)**

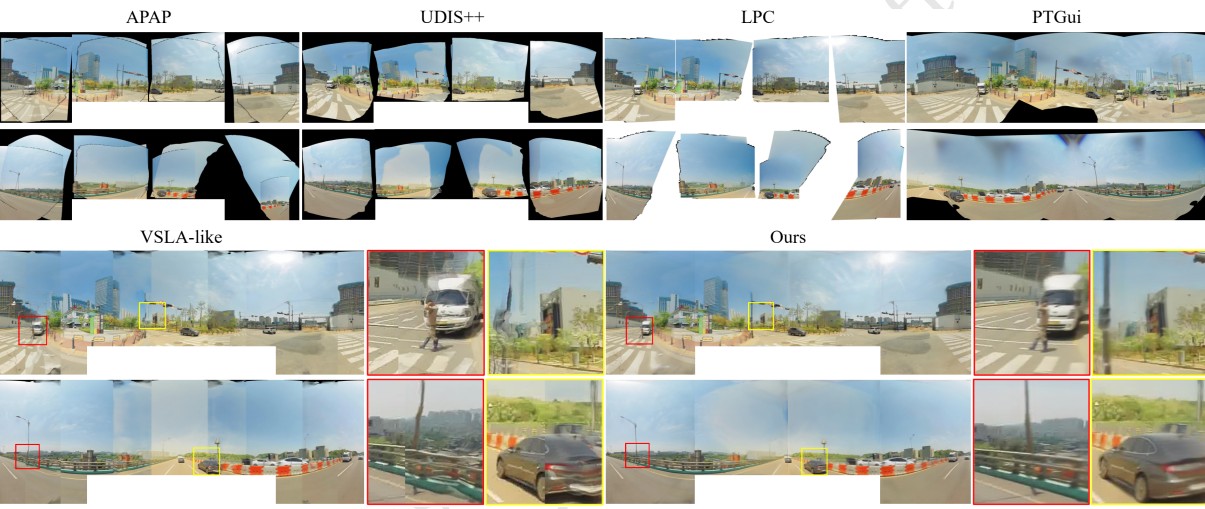

**Figure 5: Qualitative results on the real dataset in large distance parallax (1.4 m between cameras) scenario.**

similar to the Autostitch [3] method. It supports various camera inputs, but its automatic stitching often failed in the GV360 test set, particularly in instances with large distance parallax.

## 5.3 Quantitative Results

The effectiveness of our framework is evaluated by comparing it with popular image stitching methods, as outlined in 5.2. This comparison uses standard metrics such as Peak Signal-to-Noise Ratio (PSNR), Structural Similarity Index (SSIM) [53], and Learned Perceptual Image Patch Similarity (LPIPS) [62]. The comprehensive results are presented in Table 2. Our experiments span various distance parallax scenarios – 0.01 m, 0.8 m, and 1.4 m – to evaluate the adaptability of the stitching methods for general vehicle applications. Our framework, equipped with the SRM module, shows superior performance over the VSLA-like method, achieving an average 18.5% improvement of PSNR, 11.7% improvement of SSIM, and 18.0% decrease of LPIPS. It verifies the benefits of applying full region adaptation with depth-aware flow refinement and a learnable warping network in stitching multiple images, thereby enhancing model adaptability for general vehicles and improving

performance in large parallax situations. Notably, our framework shows a 14.3% decrease in LPIPS over the VSLA method in the large distance parallax scenario. This significant performance gap highlights the importance of considering depth differences and reducing flow estimation errors through additional depth information and a fine-grained warping network configuration.

## 5.4 Qualitative results

For qualitative evaluation, we compare image stitching methods on GV360 and additional unseen datasets, focusing on the presence of visual artifacts (*e.g.*, misalignment, ghosting, and duplication) and the ability to generate natural and seamless images from a human-centric perspective for omnidirectional vision. This analysis includes three different distance parallax scenarios, similar to our quantitative multi-camera VA-OVS evaluation.

*5.4.1 GV360 dataset.* The qualitative results from our dataset are illustrated in Figure 4. The results reveal that LPC and UDIS++ methods are incompatible with omnidirectional stitching due to their symmetric properties, leading to significantly inferior results.

**Table 3: Self-comparison with SRM and DAS module experiments on the full test dataset. 'P' means partial (40%) and 'M' means maximum stitching region.**

| Module | SR | IP | Cost | LFW | PSNR(↑) | SSIM(↑) | LPIPS(↓) |
|--------|-----|-----|------|-----|---------|---------|----------|
| SRM | P | | | | 23.598 | 0.792 | 0.215 |
| | M | | | | 24.891 | 0.826 | 0.236 |
| DAS | M | ✓ | | | 26.812 | 0.863 | 0.171 |
| | M | ✓ | ✓ | | 26.999 | 0.867 | 0.164 |
| | M | ✓ | | ✓ | 27.211 | 0.866 | 0.185 |
| | M | ✓ | ✓ | ✓ | **27.866** | **0.883** | **0.150** |

Commercial stitching methods such as Samsung Gear 360 and PT-Gui, although designed for 360° image stitching, struggle with large distance parallax in multi-camera settings, often resulting in duplicate and misalignment artifacts. Among these methods, VSLA-like is manually adjusted with hand-crafted image alignment and partial SR decisions to achieve more natural and seamless images. However, it is still prone to ghosting artifacts, highlighting its inability to handle depth differences. In contrast, our method effectively mitigates unintended visual artifacts and produces images that are both natural in appearance and seamlessly integrated.

*5.4.2 Unseen dataset.* We extend our comparison to include scenarios involving unseen data, which comprises images from real-world multi-camera settings, as shown in Figure 5. Despite the training of our framework on virtual samples from simulations, it has demonstrated robustness when applied to unseen datasets from real-world contexts. This result highlights our framework's generalizability to a various omnidirectional vision scenarios.

## 5.5 Ablation Study

*5.5.1 Effectiveness of Components.* In the ablation study presented in Table 3, we investigate the effectiveness of different components within our proposed modules in improving the stitching quality. First, we compare the performance of the Stitching Region Maximization (SRM) module, which aims to maximize the stitching region against a traditional method focusing only on partial stitching regions. The result shows a significant improvement in performance metrics such as PSNR and SSIM, indicating numerical and structural differences in image quality. Despite these gains, extending flow estimation to the full region may cause subtle errors, potentially leading to image-wide blurring. It can result in perceptually uncomfortable outcomes, as quantified by the LPIPS metric, highlighting a trade-off between maximizing stitching area and maintaining image sharpness.

The DAS module addresses the depth difference caused by distance parallax through three key components: the image pyramid (IP), 3D depth cost (Cost), and learnable forward warping (LFW). Notably, the absence of 3D depth costs leads to a deterioration in LPIPS scores, underscoring the pivotal role of 3D depth information in the pursuit of more natural images in omnidirectional vision applications. In addition, omitting learnable forward warping reduces PSNR and SSIM. It demonstrates the effectiveness of an integral

feature of learnable warping in detecting the importance of specific object pixels in situations with substantial depth differences between images. This enables the accurate placement of objects in the foreground, significantly enhancing the image quality.

**Table 4: Ablation study of training loss functions.**

| Training loss | PSNR(↑) | SSIM(↑) | LPIPS(↓) |
|---------------|---------|---------|----------|
| Ours | 27.866 | 0.883 | 0.150 |
| w/o $\mathcal{L}_{census}$ | 26.774 | 0.859 | 0.176 |

*5.5.2 Effectiveness of additional training loss.* We examine the effectiveness of incorporating an additional loss term, specifically the Census loss, to improve the quality of image stitching. The comparison detailed in Table 4 demonstrates the clear advantages of integrating $\mathcal{L}_{census}$ in our approach, as evidenced by the improvements across all the evaluated metrics. The PSNR and SSIM values indicate higher similarities to the ground truth images in pixel intensity and structural information. At the same time, the decrease in the LPIPS score shows a closer perceptual resemblance to the ground truth. This experiment demonstrates that including additional census loss effectively refines the output of our image stitching framework.

## 6 CONCLUSION

This paper introduces a novel omnidirectional stitching framework, referred to as OmniStitch, which addresses the challenges of dynamic distance parallax in multi-camera Vehicle-Agnostic Omnidirectional Vision Systems (VA-OVS). The framework incorporates Stitching Region Maximization (SRM) and Depth-Aware Stitching (DAS) modules to enhance stitching quality and robustness across various vehicle configurations by automating stitching region decisions and refining flow estimation with depth information to minimize artifacts. To overcome the lack of suitable datasets for VA-OVS, we provide the GV360 dataset with ground truth images that preserve 360° perspective using virtual environments. Our framework is superior in handling distance parallax variations and provides more accurate and reliable omnidirectional images. Although it currently operates with a limited number of cameras and is not completely resistant to flow estimation errors such as blurring, it provides valuable opportunities for future research to enhance the scalability and efficiency of omnidirectional vision systems for broader applications in ADAS, VR, and other fields.

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
