# OpenReview forum: "OmniStitch: Depth-Aware Stitching Framework for Omnidirectional Vision with Multiple Cameras"
_acmmm.org/ACMMM/2024/Conference — MM2024 Poster_

### Official Review · Reviewer_pbJa · 2024-05-20

**Rating:** 3
**Confidence:** 4

**Summary:**

This paper targets omnidirectional stitching in ADAS. To this end, a synthetic stitching dataset with GT and a novel stitching system is built. Compared with other methods, it performs better in the 4-camera ADAS case.

**Strengths:**

A new dataset with gt is proposed to implement high-quality stitching in 4-camera ADAS system. It covers a large diversity of scenes and camera baselines. Besides, a novel step-wise stitching pipeline is proposed.

**Limitations:**

1.	The dataset is a synthetic dataset. Although the authors conduct generalization experiments on real-world setups, the results (several qualitative results) are not convincing enough.
2.	The proposed framework seems to stitch only two images at once. It’s a little strange to me why not stitching all images once like [a][b].
3.	Since there are many unsupervised (UDIS) or weakly supervised ([a]) stitching, is it really meaningful to propose such a supervised method with a synthetic dataset?
4.	There are extensive typos in this manuscript. Please double-check this paper

Overall, I appreciate the dataset of this work. It indeed contributes to this field as there’s no such dataset. But the meaning of this dataset needs further justification. Why we need such a synthetic dataset since the unsupervised/weakly-supervised methods have already performed very well in real-world scenes. For unsupervised methods, collecting unlabeled data to learn stitching patterns under specific baselines also seems feasible.

[a] Weakly-supervised stitching network for real-world panoramic image generation
[b] End-to-end image stitching network via multi-homography estimation

**Suitability:**

3

---

### Official Review · Reviewer_PQGy · 2024-05-22

**Rating:** 4
**Confidence:** 4

**Summary:**

1. This work proposes a new panoramic stitching framework OmniStitch, which includes two main modules:

      a. Stitching Region Maximization (SRM) module: Automatically adapts to multi-camera setups for different vehicles, simplifying the image alignment and perspective conversion process.

      b. Depth-aware stitching (DAS) module: considers the large depth differences between different cameras to generate seamless panoramic images.

2. This work creates and releases a new panoramic stitching dataset, which provides accurate ground truth of 360-degree panoramic views and is used to evaluate the performance of panoramic stitching algorithms for vehicle-mounted applications.

**Strengths:**

1. A new depth-sensing panoramic stitching framework OmniStitch is proposed, which can handle the distance parallax problem between different cameras. It has great practical significance.

2. A new panoramic stitching data set is designed, which provides accurate ground truth of the 360-degree panoramic view and can be used to evaluate the performance of the panoramic stitching algorithm.

3. Experimental results show that the OmniStitch framework is better than existing panoramic stitching algorithms in handling the challenges brought by distance parallax.

In general, this work proposes a new panoramic stitching framework for the needs of vehicle-mounted panoramic vision systems, and develops corresponding data sets, which has certain innovation and practical value.

**Limitations:**

1. The data set proposed in this work can promote the development of the field, but currently we have not seen any open source information related to this data set and code. If it is open source, you can consider accepting it.

2. Although the entire network framework can systematically realize panoramic image stitching, the network innovation points are insufficient. It is recommended to design relevant network modules based on the special imaging characteristics of fisheye and panoramic images to further improve the accuracy of the method.

3. The article lacks analysis and evaluation of algorithm complexity and computational efficiency. Real-time performance is important for many applications.

4. There are few evaluations of the splicing results of real scenes, and we hope to provide more real data results.

**Suitability:**

3

---

### Official Review · Reviewer_hnGU · 2024-05-24

**Rating:** 4
**Confidence:** 4

**Summary:**

The paper focuses on ADAS and VR systems which utilize ODI camera setup. Usually in the ODI camera, multiple lenses are used to capture the image from different angles and then the images are stitched together to get final ERP image. However in case of larger vehicles, the camera might have blind spot and cannot cover most of the areas surrounding the vehicle. To overcome this, one can mount 4 different cameras at 4 corners of the vehicle and then use those 4 images to generate final ERP image, however existing algorithms will suffer due to larger parallax differences, and also dynamic nature of these camera setup which might change depending on the vehicle. To address this issues, the paper introduces a stitching framework which can work with larger depth parallax and can handle dynamic distances between the cameras involved in the capture process. The authors also opensource the dataset which can further be used to train or benchmark similar models.

**Strengths:**

**1.** The combination of the SRM and DAS modules is innovative, particularly in handling dynamic distance parallax between cameras.

**2.** The introduction and release of GV360 dataset is a major contribution. This dataset provides a wide range of dynamic distance parallax inputs and maintains a 360-degree FOV perspective. The dataset is particularly valuable for future research and development in vehicle-agnostic omnidirectional vision systems.

**3.** The adaptability to different vehicle sizes and geometries is a key strength. By eliminating the need for hand-crafted stitching regions and handling depth differences dynamically, the framework can be applied to a wide range of vehicles, enhancing its practical utility.

**Limitations:**

**1.** Since one of the target applications is to use this method on remote or autonomous vehicles where one might require real time stitching of views, the inference time of whole process is a significant measure. In the comparison tables, the inference time is missing. With its dual-module approach and incorporation of depth information, might introduce significant computational overhead. The paper does not provide a detailed analysis of the computational requirements or the potential impact on real-time performance, which is critical for applications like ADAS and autonomous driving.

**2.** The proposed framework, While the GV360 dataset is comprehensive and valuable, it is generated using a simulator. The paper would benefit from more extensive real-world validation to demonstrate the framework's effectiveness in practical scenarios. Real-world data often introduces additional complexities which are not present in simulated environments.

**3.** The framework's performance in extreme scenarios with very large depth differences or highly dynamic environments is not thoroughly explored. Additional experiments and analysis in these challenging conditions would strengthen the paper's claims about the framework's robustness.

***Other Comments***

**1.** In abstract (in the paper), line 26-28, it reads _“in addition we create and release an omnidirectional dataset, called which provides ....”_. Here the name is missing ‘GV360’.

**2.** In section 5.1 about implementation details, it will be better to mention about number of epochs. (Although iterations are mentioned, and based on batch size and number of training data one can calculate the number of epochs).

**Suitability:**

3

---

### Official Review · Reviewer_4hSE · 2024-05-24

**Rating:** 4
**Confidence:** 2

**Summary:**

The paper presents a new framework designed to create seamless 360-degree panoramas from multiple cameras, specifically for applications like Advanced Driver Assistance Systems and Virtual Reality. The OmniStitch framework includes two key components: the Stitching Region Maximization (SRM) Module for adapting stitching regions dynamically across different vehicle types, and the Depth-Aware Stitching (DAS) Module, which minimizes errors like ghosting by addressing depth disparities. The paper also introduces a new dataset tailored for omnidirectional systems, demonstrating that OmniStitch outperforms existing methods in generating high-quality panoramic images.

**Strengths:**

The evaluation of the OmniStitch framework is thorough, with comprehensive testing using the newly introduced General-purpose Vehicle’s 360-degree (GV360) dataset. This dataset is specifically designed to simulate real-world conditions for vehicle-agnostic systems. The paper provides comparative analyses showing that OmniStitch outperforms existing state-of-the-art methods in terms of image quality metrics such as PSNR, SSIM, and LPIPS. This extensive evaluation demonstrates the effectiveness of the proposed methods under various conditions;
The paper highlights significant applications of the OmniStitch framework, particularly in areas requiring high-quality panoramic images such as ADAS and VR. By improving the adaptability and quality of image stitching, the framework has the potential to enhance the development of systems in these fields, which require robust and reliable visual information from multiple cameras.

**Limitations:**

Although the workings of the SRM and DAS modules are detailed, the paper lacks an in-depth analysis of the algorithm's complexity and runtime. Besides, I am wondering if you have consider the cyclic consistency inside an ERP image(the output of the Framework) which is a good property for stitching. For real-time applications, such as advanced driver-assistance systems, processing speed is critical. The real-time performance and resource consumption of the algorithm are not explicitly discussed, which could be a potential flaw. To meet the expected length for a conference paper, I suggest including more detailed explanations and additional analysis in your manuscript.

**Suitability:**

3

---

### Meta-Review · Area_Chair_eD9n · 2024-07-04

**Recommendation:** Accept (Poster)
**Confidence:** 4

**Metareview:**

The paper is written in a clear manner and provides an original contribution to the field, as indicated by almost all of the 4 reviewers.
The main contributions are the new panoramic stitching framework OmniStitch with its two modules Stitching Region Maximization (SRM)  and Depth-aware stitching (DAS), the synthetic dataset DV360 and a rather comprehensive evaluation of the approach.
Based on the reviews, the authors have provided a rebuttal that mostly addresses all points of criticism.
Nevertheless, the overall review score has slightly decreased, as one reviewer (R4) stated that his concerns were not all addressed (but also stated that some were actually addressed). In turn, one reviewer did not modify his / her rating but indicated that he / she would now lean toward acceptance as a key criticism was resolved.
3 out of 4 reviewers agree in their rating providing a "borderline accept", while 1 reviewer now recommends "weak reject" (initially "borderline reject"). From the 3 reviewers that agree, 2 are "very confident" in their vote, as is the reviewer now suggesting "weak reject".

Besides the aforementioned strong points, the following aspects were found positive:
The contribution is specifically relevant for the parallax-induced issues in case of long and dynamic distances between cameras as for long vehicles, as appreciated by some reviewers (R2, R3).
Here, also the dataset represents a specific contribution (all reviewers).

Further negative points mentioned are:
The details desired by some reviewers about runtime and complexity have mostly been addressed in rebuttal, as mentioned by the respective reviewers (R1, 2, 3).
The real-world validation requested by some reviewers (R2, 3, 4) has partially been addressed in rebuttal, but must be more comprehensively covered in case of acceptance of the paper, by including and extending the information from the rebuttal in the camera ready (CR) paper.

Although some of the points by the critical review R4 remain valid, I am not fully agreeing with conclusions such as that datasets are not needed because there already are datasets or that models are not needed because there are certain other models, in case that certain advantages such as increased training data or improved performance may result. If of course neither in application nor performance new datasets or models are meaningful contributions, the merit of the work may be limited. The three other reviewers see this differently.
The overall rating is 3.75. Hence, in case that no sufficient space is available, this paper may also be rejected. However, as the aforementioned detailed points may be best discussed with peers at a conference, in summary, I lean towards accepting the paper as a poster. Here, direct communication with others working on related topics may enable fruitful exchanges and further improvements of existing algorithms.